# Artificial Intelligence and Machine Learning Applications to Pharmacokinetic Modeling and Dose Prediction of Antibiotics: A Scoping Review

**DOI:** 10.3390/antibiotics13121203

**Published:** 2024-12-10

**Authors:** Iria Varela-Rey, Enrique Bandín-Vilar, Francisco José Toja-Camba, Antonio Cañizo-Outeiriño, Francisco Cajade-Pascual, Marcos Ortega-Hortas, Víctor Mangas-Sanjuan, Miguel González-Barcia, Irene Zarra-Ferro, Cristina Mondelo-García, Anxo Fernández-Ferreiro

**Affiliations:** 1Clinical Pharmacology Group, Health Research Institute of Santiago de Compostela (IDIS), 15706 Santiago de Compostela, Spain; iriavarela13@gmail.com (I.V.-R.); enriquebandinvilar@gmail.com (E.B.-V.); kikotoja@gmail.com (F.J.T.-C.); antonio.canizo.outeirino@sergas.es (A.C.-O.); francisco.cajade.pascual@sergas.es (F.C.-P.); miguel.gonzalez.barcia@sergas.es (M.G.-B.); irene.zarra.ferro@sergas.es (I.Z.-F.); 2Pharmacy Department, University Clinical Hospital of Santiago de Compostela (SERGAS), 15706 Santiago de Compostela, Spain; 3Pharmacology, Pharmacy and Pharmaceutical Technology Department, Faculty of Pharmacy, University of Santiago de Compostela (USC), 15782 Santiago de Compostela, Spain; 4VARPA Group, INIBIC, Research Center CITIC, University of A Coruña, 15071 A Coruña, Spain; m.ortega@udc.es; 5Department of Pharmacy and Pharmaceutical Technology and Parasitology, University of Valencia, 46010 Valencia, Spain; victor.mangas@uv.es; 6Interuniversity Research Institute for Molecular Recognition and Technological Development, Polytechnic University of Valencia, 46010 Valencia, Spain

**Keywords:** pharmacokinetics, artificial intelligence, machine learning, therapeutic drug monitoring, antibiotics

## Abstract

**Background and Objectives:** The use of artificial intelligence (AI) and, in particular, machine learning (ML) techniques is growing rapidly in the healthcare field. Their application in pharmacokinetics is of potential interest due to the need to relate enormous amounts of data and to the more efficient development of new predictive dose models. The development of pharmacokinetic models based on these techniques simplifies the process, reduces time, and allows more factors to be considered than with classical methods, and is therefore of special interest in the pharmacokinetic monitoring of antibiotics. This review aims to describe the studies that use AI, mainly oriented to ML techniques, for dose prediction and analyze their results in comparison with the results obtained by classical methods. Furthermore, in the review, the techniques employed and the metrics to evaluate the precision are described to improve the compression of the results. **Methods**: A systematic search was carried out in the EMBASE, OVID, and PubMed databases and the results obtained were analyzed in detail. **Results**: Of the 13 articles selected, 10 were published in the last three years. Vancomycin was monitored in seven and none of the studies were performed on new antibiotics. The most used techniques were XGBoost and neural networks. Comparisons were conducted in most cases against population pharmacokinetic models. **Conclusions**: AI techniques offer promising results. However, the diversity in terms of the statistical metrics used and the low power of some of the articles make the overall assessment difficult. For now, AI-based ML techniques should be used in addition to classical population pharmacokinetic models in clinical practice.

## 1. Introduction

Artificial intelligence (AI) is a rapidly growing and highly useful area when working with large amounts of data. For this reason, it is being rapidly developed and implemented to automate decision-making in health systems [1,2]. The digital transformation in healthcare has been promoted in recent years by global institutions [3].

Automatic learning techniques, also known as machine learning (ML), constitute a field of statistical research within AI whose objective is the formation of computational algorithms that divide, classify, and transform a dataset to maximize the capacity to classify, predict, group, or discover patterns.

In the area of pharmacology, the application of ML is being explored in several fields: pharmacokinetics/pharmacodynamics (PK/PD) and dose optimization [4,5], the relationship between structure and pharmacological activity [6,7,8], prediction of adverse effects [9,10,11], prediction of drug interactions [12,13] and simulation of clinical trials [14,15]. The most explored field so far has been the relationship between structure and biological activity. However, the area of PK/PD and dose optimization is progressing [16].

The use of ML in PK is mainly focused on pharmacokinetic modeling and dose adjustment based on this approach [2,17]. The primary purpose of studying the PK/PD characteristics of a biologically active product is to deduce relationships between measurable variables (e.g., drug concentration) and the response [18]. The optimization of the model is often based on an accumulation of trial–error tests through which changes are made and adjusted again until it is optimized [19]. This is a perfect scenario for the implementation of AI techniques, with the aim of automating the iterative process of building the model. Growing access to big data in recent years has increased interest in using ML directly for predictions. Starting from large databases from which the ML learns, it is able to establish relationships between different parameters that can influence the variability of the observed concentration and estimate it correctly [20].

ML has the advantage of being fast, efficient, and able to handle large datasets so it can be used to identify complex relationships [21]. This is of particular interest for the development of predictive dose models for newly developed drugs [22]. On the other hand, physiologically based biopharmaceutics models (PBPK) are based on biological mechanisms, facilitating mechanistic understanding, biological interpretability of results, and the ability to conduct in silico experiments using model simulations. In this sense, ML carries the inherent risk of producing clinically irrelevant results. For all these reasons, it seems that the key is to combine these two disciplines and work together, taking advantage of the benefits that each of them offers [23].

PK is especially useful and necessary for the therapeutic monitoring of antimicrobial drugs, in which effectiveness and toxicity depend, in most cases, on the concentration of the drug [24,25,26]. Due to the widespread emergence of antimicrobial resistance, the optimal use of both common and newly developed antibiotics is of vital importance [27,28,29]. With the aim of reducing the appearance of resistance and extending its useful life, pharmacokinetic monitoring is a tool that should be used in clinical practice whenever possible. The need for rapid development of new antibiotics means that, in many cases, it is not possible to develop pharmacokinetic models of all of them in time, so ML techniques would have special relevance in this field [30,31].

In this review, we aim to analyze the evidence for the use of AI and ML techniques in the pharmacokinetic modeling and therapeutic drug monitoring of antimicrobials as well as evaluate the accuracy and precision in comparison with conventional monitoring techniques. This is the first review carried out on this topic and, consequently, it will help to understand the current paradigm of the use of these techniques in the pharmacokinetic model of antibiotics and clarify their potential.

## 2. Results

The PRISMA flowchart of the review is shown in Figure 1. In total, 205 publications were identified. After removing duplicates and screening by title/abstract, 183 publications were excluded and 22 were selected for full-text screening. Of these, 3 of them were eliminated because they were not in full text and 5 were subsequently excluded after retrieval, leaving 13 studies for evaluation. All of them were original articles.

### 2.1. AI and ML Techniques Used for PK Modeling and Dose Prediction

In the studies selected and described in this review, a wide variety of ML techniques are used, and, in most cases, authors test several of them and then choose the one that performs best or combine them to obtain an ensemble method. In order to clarify and adequately understand the paradigm of the use of AI and ML techniques in the pharmacokinetics of antibiotics, which is the objective of this review, the techniques used are classified and explained below.

First, we need to know that the term AI covers a variety of technologies, methods, and algorithms, focusing on different applications. In the field of PK, ML algorithms have been the most studied ones [32]. ML techniques are divided into three main types: supervised learning, unsupervised learning, and reinforcement learning. In supervised learning, both the input (e.g., drug dose) and output (e.g., plasma concentration) are known, and the algorithm searches for the mathematical function that best describes the relationship between the two. Unsupervised learning techniques are most applicable when finding subgroups or clusters in large datasets (e.g., patients with a higher probability of developing a certain adverse effect) [33]. Reinforcement learning is an ML training method based on rewarding desired behaviors and punishing unwanted ones. In pharmacometrics, supervised learning algorithms are the most commonly used [16,34].

Supervised learning algorithms include regression such as linear, logistic, lasso, ridge, and elastic nets, random forests, support vector machines, k-nearest neighbors, decision trees, naïve Bayes, neural networks, and gradient boosting algorithms [33,35]. The classification of algorithms is not clear and differs greatly depending on the literature. Furthermore, in many cases, the same algorithm can be used for both regression and classification, and in others, the final technique used is often the result of a combination of several algorithms. The ML algorithms most used in pharmacokinetic analysis and those that have been used in the articles selected in this review are classified in Figure 2 and explained in Appendix A.

### 2.2. Evaluation of the Available Data

The selected articles in this review and all the analyzed parameters of each one are depicted in Table 1. Of the 13 articles finally selected, 7 focus on vancomycin dose prediction [36,37,38,39,40,41,42]. Aminoglycosides are the second most used group of antibiotics to test AI techniques, with three in total (one gentamicin, one arbekacin, and one tobramycin) [43,44,45]. Beta-lactams are used in two articles (one using piperacillin and another in which cefepime, azlocillin, ceftazidime, and amoxicillin are used) [46,47]. Finally, another article uses rifampicin as an example to evaluate algorithms developed using ML [23].

It is important to highlight that, despite the novelty of the use of AI techniques in pharmacometrics and their special interest in reducing the development times of population PK models for the new antibiotics developed, these techniques are currently being tested with the antibiotics for which there is more experience in monitoring by typical pharmacokinetic models, such as vancomycin and aminoglycosides. This fact shows that the use of these techniques is still in the preliminary phase and needs to be compared and validated against conventional pharmacokinetic monitoring techniques.

Another notable aspect is the growing interest in this topic in recent years, with 10 of the 13 articles analyzed published in the last 3 years. This exponential growth is in line with the growing interest in the application of artificial intelligence in health systems [1,48] and, also, in the field of clinical PK [35,49].

The studies analyzed that evaluate the capacity of these new techniques in pharmacometrics are studies with a retrospective design based on large volumes of data. Most are carried out on real patients, although some studies are realized on mathematical simulations of patients, either for the development of the method or for its validation [23,36,37]. In most studies, these databases are created from medical records from different hospitals. There are differences in the types of populations studied. In four of the selected articles, the study population is adult patients who are being treated with the selected antibiotic, without any other specific parameter. A total of three works focus on more complex patients admitted to intensive care units (critical, burned, or post-surgical patients). One of the articles includes exclusively patients with tuberculosis. It is notable that in five of the selected articles, the study population included pediatric patients, three of them neonates (see Table 2). This last point is especially interesting since one of the postulates of the use of AI and ML techniques in pharmacometrics is the ability to estimate pharmacokinetic parameters a priori (without having previous samples), an aspect that is especially interesting in this subpopulation due to the difficulty for sampling [50,51].

Another relevant aspect when analyzing this type of study is the type of variables included in the model. One of the most promising aspects of using ML techniques is their ability to include large types of variables and study their possible effect on pharmacometrics. In the selected articles, some novel clinical variables are included such as the type of infection [45], medications administered during the last year [38], percentage of burned area [44], or numerous analytical parameters that are not common to use in conventional pharmacokinetic models such as albumin, bilirubin, fluid balance, complete blood counts, or inflammation parameters [42,46]. However, most studies include more common covariates such as weight, height, race, age, or creatinine clearance. The most used AI techniques were approaches such as XGBoost (5/13) and neural networks (NNs) (5/13), followed by decision or regression trees (DTs or RTs) (4/13), or others such as random forest (RF) (3/13) or lasso (2/13). In many of the articles, several techniques are used in the same model.

Many works initially used several methods and finally selected the one that allowed for better prediction. In others, several approaches are combined to form “ensemble models”. In most of the articles, the new models are compared with classic population PK models (11/13), with seven of them in NONMEM. In other cases, they are compared against previously published nomograms [40] or by multivariate logistic regression [44].

The metrics used to measure the precision of AI techniques are highly variable throughout the studies analyzed (see Table 2) and not all are used for external validation and comparison with the conventional pharmacokinetic model. Therefore, it is difficult to realize a global evaluation of the results of these techniques compared to conventional models, which makes it impossible to reach a clear result. This limitation had already been previously postulated by Mehdi et al. [52]. However, by analyzing the studies one by one, quite acceptable prediction error values are obtained, consistent with those obtained by conventional methods. In many of them, a numerical reduction in the errors is observed compared to the usual models, without being able to say that the improvement is significant. In all articles, the authors conclude that AI techniques are one more tool that can help improve the prediction of pharmacokinetic parameters and dose prediction, positioning itself for the moment as a complementary technique that can be very useful, especially for pharmacometricians with little experience or when a faster analysis is required. The main limitations found in the studies are that some use very small sample sizes and that ML techniques have worse predictions than conventional methods when it comes to values outside the usual range.

**Table 1 antibiotics-13-01203-t001:** Articles included. ATB: antibiotic; N: number of sample; ML: machine learning; XGBoost: extreme gradient boosting; RF: random forest; GBM: gradient boosting machine; LASSO: lasso regression; BMI: body mass index; s-Cr: serum creatinine; CrCl: creatinine clearance; LightGBM: light gradient boosting machine; GLMNET: generalized linear model via penalized maximum likelihood; MARS: multivariate adaptative regression splines; KNN: K-nearest neighbors; DT: decision tree; Adaboost: adaptive boosting; ETR: extra tree regressor; GBR: gradient boosting regression (with ridge, lasso, or elastic net regularization); CV: cardiovascular; MAA: model averaging approach; MSA: model-selection algorithm; NNs: neural networks; GBT: gradient boosting trees; GP: gaussian processes; MLP: multilayer perceptron model; SVR: support vector regression; PCR: C-reactive protein; CART: classification and regression tree; AST: aspartate aminotransferase; ALT: alanine aminotransferase; GBDT: boosting decision tree; CatBoost: categorical boosting; LBHM: light gradient boosting machine; LR: logistic regression; Tabnet: tabular neural network; ANN: approximate nearest neighbor. popPK: population pharmacokinetic model. * Metrics’ abbreviations explained in Table 2. ** PAR: “percentage of dose in the acceptable range” is not a common metric, as defined by the authors of the article.

Study	ATB	N	Subject Characteristics	Objective	AI Technique Used	Clinical Parameters Involved	Model Compared	Precision Metrics *	Results	Remarks and Conclusions
Keutzer et al., 2022 [23]	Rifampicin	1826 simulations	Tuberculosis patients	To examine the ability of various ML algorithms to predict time-varying plasma concentrations and derive PK parameters.	XGBoost, RF, GBM, LASSO	Age, BMI, dose, fat-free mass, infection by VIH, VIH coinfection, height, treatment week, race, gender, time after dose, weight.	popPK model in NONMEM	R^2^, RMSE, MAE	-For concentration prediction, the best AI technique was XGBoost using 6 rifampicin concentrations (R^2^ = 0.84, RMSE = 6.9 mg/L, MAE = 4 mg/L. -For AUC_0–2_ prediction, the best AI technique was LASSO using 6 rifampicin concentrations (R^2^ = 0.97, RMSE: 29.1 h.mg/L, MAE: 18.8 h.mg/L)	Prediction was according to the PK model. AI was 22 times faster.
Wang et al., 2022 [38]	Vancomycin	2282 real patients	Patients who received at least one vancomycin injection	To develop an innovative method to suggest the initial and subsequent daily dose of vancomycin.	LightGBM	s-Cr, CrCl, age, weight, gender, age, albumin, medicines on CV system, on alimentary tract or metabolism, on blood-forming organs, hemodialysis, daily dose and concentration, administration timing.	popPK model	MAE, PAR **	*-Initial dose model:* MAE: 450.2 mg/day (AI model) vs. 727.5 mg/day (popPK model); PAR: 51.7% (AI model) vs. 28.3% (popPK model) *-Subsequent dose model:* MAE: 267.1 mg/day (AI model) vs. 392.1 mg/day (popPK model); PAR: 73.4% (AI model) vs. 60.4% (popPK model)	ML performed better than the popPK model.
Bououda et al., 2022 [36]	Vancomycin	28 real patients/6000 simulations	Obese, critically ill, hospitalized patients with sepsis, trauma, and post-heart surgery	To train an ML algorithm to predict vancomycin AUC from early concentrations and few features and predict vancomycin AUC from early concentrations.	XGBoost	Concentration, s-Cr, age, weight, height, gender, dose, administration timing.	popPK models: MAA, MSA, PKJust program	rMPE, rRMSE	rMPE 0.97 and rRMSE 12.7% when compared with PKJust; rMPE 0.8 and rRMSE 11.9% when compared with MSA; rMPE 1.2 and rRMPE 11.8% with MAA.	XGBoost algorithms seem complementary to standard popPK approaches.
Ponthier et al., 2022 [37]	Vancomycin	82 real patients/1900 simulations	Term and preterm neonates	To obtain an ML algorithm to estimate the best vancomycin initial dose and compare it to a previously validated popPK model.	XGBoost, GLMNET, MARS	Gestational age, time of infection after birth, postmenstrual age at first infection, current weight, s-Cr.	popPK model	rMPE, rRMSE	XGBoost was the best. In the training set, rRMSE was 36.1 and rMPE was 7.2 in the train set. In the test set, rRMSE was 35.7 and rMPE was 8.6. The numerical best target attainment rate was obtained with the ML algorithm (35.3% vs. 28%).	ML algorithm improves the exposure target attainment rate.
Tang et al., 2021 [47]	Vancomycin, latamofex, cefepime, azlocillin, ceftazidime, and amoxicillin	2272 real patients	Neonates	To evaluate whether the combination of ML methods and popPK methods can accurately predict individual clearance of renally eliminated drugs.	KNN, DT, adaboost, ETR, RF, GBR	Birth weight, current weight, gestational age, postnatal age, postmenstrual age, s-Cr.	popPK model in NONMEM	R^2^, MSE, MRE%	ETR was selected as the final uniform ML approach. Combined predictive method (popPK + AI) had an MRE of 15.4%, 2.2%, 2.8%, 10.1%, and 2% for vancomycin, cefepime, latamoxef, azlocillin amoxicillin, and ceftazidime, respectively. Except for azlocillin (9.9% of MRE in the popPK model), all the MREs were lower with the combined method.	The combination of popPK and the machine learning approach provided consistent information.
Brier et al., 1995 [43]	Gentamicin	144 real patients	Patients who received gentamicin in the service of the Veterans Administration Medical Center in Louisville	To use a neural network to predict peak and trough gentamicin concentrations and compare the results with the NONMEM model.	NN	Age, height, weight, s-Cr, CrCl, dose, dose/weight, dose interval, BMI.	popPK model in NONMEM	PE, PE%, SPE, AE, APE (or AE%)	-Gentamicin’s peak: PE was −0.02 in NN vs. 0.14 in NONMEN, PE% was −2.45% in NN vs. 1.04% in NONMEN, SPE was 0.67 in NN vs. 0.83 in NONMEN, and APE was 16.5% in NN vs. 18.6% in NONMEN. -Gentamicin´s trough: PE was 0.002 in NN vs. 0.049 in NONMEN, PE% was −11.11% in NN vs. −14.5% in NONMEN, SPE was 0.58 in NN vs. 0.67 in NONMEN, and APE was 48.3% in NN vs. 59% in NONMEN. NONMEM was more precise in the prediction of concentration outside the range 2.5–6 µg/mL (*p* = 0.098)	NNs perform well when they are used in the range of concentrations that they have trained. An NN has limitations in predicting out-of-range concentrations.
Verhaeghe et al., 2022 [46]	Piperacillin	282 real patients	Surgical critically ill patients treated with piperacillin/tazobactam in continuous infusion	To use the ML model to predict total plasma concentrations of piperacillin in critically ill patients a priori and a posteriori and compare the results with a popPK model.	GBT, GP, MLP	Piperacillin previous concentrations, sex, weight, s-Cr, CrCl, albumin, bilirubin, fluid balance, height, lactate, platelets, red blood cells, sex, hours since start of treatment.	popPK model in NONMEM	PE, MAE, RMSE, R^2^, MdAPE, MdPE	-A priori method: RMSE (GBT 34.27; GP 37.41; MLP 38.56; PK 57.97), MAE (GBT 21.55; GP 23.54; MLP 27.35; PK 39.67), ME (GBT −4.09; GP 2.04; MLP 2.58; PL −30.27), MdAPE (GBT 17.29%; GP 21.39%; MLP 23.09%; PL 40.79%), MdPE (GBT 0.06%; GP −3.83%; MLP −5.34%; PK 38.33%) -A posteriori method: RMSE (GBT 32.93; GP 34.03; MLP 37.20; PK 49.58), MAE (GBT 18.22; GP 19.41; MLP 23.64; PK 31.28), ME (GBT −6.55; GP −3.83; MLP −4.87; PK 4.91), MdAPE (GBT 12.75%; GP 16.48%; MLP 17.06%; PK 26.09%), MdPE (GBT 1.77%; GP −376%; MLP 0.73%; PK −1.85%)	ML models can consistently estimate piperacillin concentrations with high predictive accuracy, especially in a priori methods.
Huang et al., 2021 [42]	Vancomycin	407 real patients	Pediatric patients who received vancomycin intravenously	To establish an optimal model to predict vancomycin through concentrations in pediatric patients by using ML.	DT, SVR, RF, Adaboost, Bagging, ETR, GBRT, XGBoost; then, the best five were selected and an “ensemble model” was performed	Vancomycin dose concentrations and intervals, age, height, weight, gender, CrCl, uric acid, procalcitonin, PCR, AST, ALT, bilirubin, complete hemogram.	popPK model in NONMEM	R^2^, MSE, RMSE, MAE	The results of ML were superior to the popPK model. Ensemble model: MSE 34.39, R2 0.614, MAE 3.32, RMSE 4.94. The accuracy of the predicted through concentration (±30%) was 51.22% in the ML model vs. 36.59% in the popPK model.	The results of ML are better than the popPK model in predicting vancomycin concentration.
Yamamura et al., 2004 [44]	Arbekacin (aminoglycoside used in Japan)	30 real patients	Burn patients hospitalized in an intensive care unit in Japan	Use artificial neural network modeling to predict arbekacin plasma concentration and compare with logistic regression analysis.	NN	Dose, parenteral fluid, BMI, CrCl, burn area after operation.	Multivariate logistic regression models	R^2^	Artificial neural networks had an r of 0.9862 and logistic regression had an r of 0.8829.	Artificial neural networks were superior compared with logistic regression analysis.
Tang et al., 2023 [41]	Vancomycin	1631 real patients	Neonates and young infants with postmenstrual age ranging from 23.3 to 52.4 weeks	To assess whether ML can be used in clinical practice to predict treatment targets and calculate optimal dosing regimens for individual patients.	GBDT, CatBoost, XGBoost, LBHM, LR, SVR, Tabnet, ANN	s-Cr, sampling time, single dose per unit body weight, frequency of dosing within 24 h, postnatal age, vancomycin concentration assay method, gestational age at birth, birth weight.	popPK model	RMSE, R^2^, MAPE (or MAE%), MPE (or ME%)	CatBoost was the optimal ML method. For C_min_ prediction, RMSE was 5.02 for the ML model vs. 6.18 for popPK; MAPE was 29.5% for the ML model vs. 53% for popPK and MPE was −4.20% for ML vs. 12.4 for popPK.	The ML model was developed to be accurate and precise and can be used for individual dose recommendations in neonates.
Chow et al., 1997 [45]	Tobramycin	101 real patients	Pediatric patients who received tobramycin intravenously in Tucson	To explore the applicability of the neural network approach to capture the relationship between patient-related prognostic factors and plasma drug levels.	NN: test I (with accumulated times), test II (without accumulated times)	Age, weight, gender, illness, dose, dosing interval, time of blood drawn.	popPK model in NONMEM	MSE, ME, ME% (or MPE), AE% (or APE)	Test I turned out worse than NONMEN. Test II provided precision of the predicted concentrations comparable to that of the NONMEN analysis: MSE 1.88 NONMEN vs. 1.78 NN. AE% was better for NN test II: 33.9% vs. 39.9% in NONMEN. ME was smaller in NONMEN (0.077 vs. 0.32); PE% was better in NN (2.59% vs. 17.3% in NONMEN)	NNs could capture the relationships between patient-related factors and plasma drug levels.
Nigo et al., 2022 [39]	Vancomycin	5483 real patients	Adults who had at least one serum vancomycin level after their first vancomycin dose; patients with ECMO, hemodialysis, and renal replacement therapy were excluded.	To develop a new PK approach with RNN-based methods with electronic medical records (EHRs) to achieve more accurate and individualized predictions for vancomycin serum concentration in hospitalized patients.	NN	Weight, height, vital signs, laboratory biochemistry and complete hemogram, vancomycin dose and previous concentration, concomitant medications.	popPK model in NONMEM (VTDM model)	RMSE, MAPE (or MAE%), MAE	PK-RNN-V E vs. VTDM: RMSE 5.39 vs. 6.29; MAE 3.64 vs. 4.26; MAPE 25.41% vs. 29.15%.	PK-RNN-V E exhibits better RMSE, MAE, and MAPE compared to any of the VTDM models. PK-RNN-V E can integrate real-time patient-specific data from an EHR.
Miyai et al., 2022 [40]	Vancomycin	822 real patients	Patients who received vancomycin intravenously and had the concentration measured at least once	To construct a model for estimating the vancomycin maintenance dose to achieve the target.	CART	Age, BMI, CrCl.	Other nomograms (Oda et al. [53], Thomson et al. [54])	ME% (or MPE), MAE% (or MAPE)	DT model: ME 10%, MAE 26.7%; nomogram, Oda et al. [53]: ME 0.77%, MAE 26.6%; nomogram, Thomson et al. [54]: ME 8.67%, MAE 26.5%.	The constructed model can help construct clinical models for dose setting of initial vancomycin administration.

**Table 2 antibiotics-13-01203-t002:** Statistical metrics to evaluate the prediction accuracy of techniques employed. ${\hat{y}}_{i}=\text{predicted}\;\text{value}$; $y_{i}=\text{observed}\;\text{value}$.

Metric	Type of Statistical Metric	Type of Evaluation	Definition	Formula	Units	Interpretation
Coefficient of determination (R-squared) R^2^	regression metric	accuracy or bias	proportion of the total variance of the variable explained by the regression	$R^{2}=1-\frac{\sum {\left(y_{i}-{\hat{y}}_{i}\right)}^{2}}{\sum {\left(y_{i}-{\bar{y}}_{i}\right)}^{2}}$	no units	Represents the proportion of the variance in the dependent variable which is explained by the linear regression model. Values between 0 and 1, where 1 is the best value.
Prediction error (PE)	difference	accuracy or bias	difference between the predicted and observed concentrations	$\mathrm{PE}={\hat{y}}_{i}-{\;y}_{i}$	concentration units	Values between 0 and ∞. A lower PE indicates superior model accuracy.
Prediction error percentage (PE%)	percentage metric	accuracy or bias	percentage of PE	$\mathrm{PE}\;\left(\%\right)=\frac{\left({\hat{y}}_{i}-y_{i}\right)}{\left(y_{i}\times 100\right)}$	%	Values between 0 and 100. A lower PE% indicates superior model accuracy
Absolute prediction error (AE)	difference	accuracy or bias	difference between the predicted and observatory concentrations in absolute values	$\mathrm{APE}=\left|{\hat{y}}_{i}-{\;y}_{i}\right|$	concentration units	Values between 0 and ∞. A lower APE indicates superior model accuracy.
Absolute prediction error percentage (APE or AE%)	percentage metric	accuracy or bias	percentage of APE	$\mathrm{APE}\;\left(\%\right)=\frac{\left|\left({\hat{y}}_{i}-y_{i}\right)\right|}{\left(y_{i}\times 100\right)}$	%	Values between 0 and 100. A lower APE% indicates superior model accuracy.
Mean prediction error (ME)	median of the difference	accuracy or bias	sum of prediction errors divided by the sample size	$\mathrm{ME}=\frac{\sum \left({\hat{y}}_{i}-{\;y}_{i}\right)}{N}$	concentration units	Values between 0 and ∞. A lower MPE indicates superior model accuracy.
Mean prediction error percentage (MPE or ME%)	percentage metric	accuracy or bias	percentage of MPE	$\mathrm{MPE}\;\left(\%\right)=\frac{\sum \left[\left({\hat{y}}_{i}-y_{i}\right)/\left(y_{i}\times 100\right)\right]}{N}$	%	Values between 0 and 100. A lower MPE% indicates superior model accuracy.
Mean absolute prediction error (MAE)	median of the difference	precision	sum of absolute errors divided by the sample size.	$\mathrm{MAE}=\frac{\sum \left(\left|{\hat{y}}_{i}-{\;y}_{i}\right|\right)}{N}$	concentration units	Values between 0 and ∞. A lower MAE indicates superior model accuracy.
Mean absolute prediction error percentage (MAPE or MAE%)	percentage metric	precision	percentage of MAPE	$\mathrm{MAPE}\;\left(\%\right)=\frac{\sum \left[\left|\left({\hat{y}}_{i}-y_{i}\right)\right|/\left(y_{i}\times 100\right)\right]}{N}$	%	Values between 0 and 100. A lower MAPE% indicates superior model accuracy.
Root mean squared error (RMSE)	squared root	precision	quantifies the differences between predicted values and actual values, squaring the errors, taking the mean, and then finding the square root; computed by taking the square root of MSE	$\mathrm{RMSE}=\sqrt{\frac{\sum {\left({\hat{y}}_{i}-{\;y}_{i}\right)}^{2}}{N}}$	concentration units	Values between 0 and ∞. Lower values indicating better predictive accuracy.
Root mean squared error percentage (RMSE%)	squared root relative	precision	percentage of the RMSE	$\mathrm{RMSE}\;\left(\%\right)=\sqrt{\frac{\sum {\left[\left({\hat{y}}_{i}-y_{i}\right)/y_{i}\right]}^{2}}{N}}$	%	Values between 0 and 100. A lower RMSE% indicates superior model accuracy.
Median prediction error percentage (MDPE%)	median	precision	MDPE is found by ordering PE from smallest to largest, and using this middle value	$\text{MDPE}\;\left(\%\right)=\text{median}\;\text{PE}\;\left(\%\right)$	%	Values between 0 and 100. A lower MDPE indicates superior model accuracy.
Median absolute prediction error percentage (MDAPE%)	median	precision	MDAPE is found by ordering the APE from smallest to largest and using this middle value	$\text{MDAPE}\;\left(\%\right)=\text{median}\;\text{APE}\;\left(\%\right)$	concentration units	Values between 0 and ∞. A lower MDAPE indicates superior model accuracy.
Mean relative error (MRE)	relative ratio	accuracy or bias	MRE is defined as the ratio between MAPE and the reference E-field magnitude within the corresponding target region	$\mathrm{MRE}=\frac{\sum \left|\left({\hat{y}}_{i}-y_{i}\right)\right|}{\left({\hat{y}}_{i}/N\right)}$	no units	Values between 0 and 1
Mean relative error percentage (MRE% or rMPE)	relative percentage	accuracy or bias	percentage of MRE	$\mathrm{MRE}\;\left(\%\right)=100\;\frac{\sum \left|\left({\hat{y}}_{i}-y_{i}\right)\right|}{\left({\hat{y}}_{i}/N\right)}$	%	Values between 0 and 100. A lower MRE% indicates superior model accuracy.
Square prediction error (SPE)	square of the difference	precision	measures the expected squared distance between what your predictor predicts for a specific value and what the true value is	$\mathrm{SPE}={\left({\hat{y}}_{i}-y_{i}\right)}^{2}$	square concentration units	Values between 0 and ∞. A lower SPE indicates a better model.
Mean square error (MSE)	mean of the difference	precision	squaring the difference between the predicted value and actual value and averaging it across the dataset	$\mathrm{MSE}=\frac{\sum {\left({\hat{y}}_{i}-y_{i}\right)}^{2}}{N}$	concentration units	Values between 0 and ∞. MSE increases exponentially with an increase in error. A good model will have an MSE value closer to zero.
Relative mean prediction error (RMPE or rMPE)	relative percentage	accuracy or bias	a variant of root MPE, gauging predictive model accuracy relative to the target variable range	$\mathrm{rMRE}=100\times \frac{\sum \left[{\hat{y}}_{i}-\left({\hat{y}}_{i}/N\right)\right]}{\left({\hat{y}}_{i}/N\right)}$	%	Value between 0 and 100. Lower rMPE shows lower deviation.
Relative root mean squared error (RRMSE or rRMSE)	relative percentage	accuracy or bias	a variant of RMSE, gauging predictive model accuracy relative to the target variable range	$\mathrm{rRMSE}\;\left(\%\right)=\sqrt{\frac{\sum \left\{{\left[\left({\hat{y}}_{i}-y_{i}\right)/y_{i}\right]}^{2}/N\right\}}{\left({\hat{y}}_{i}/N\right)}}$	%	Value between 0 and 100. <10% is an excellent value for RRMSE. >30% is a poor value for RRMSE.

### 2.3. Quality Assessment on Prediction Accuracy of Techniques Employed

To compare the prediction capacity of a pharmacokinetic method based on AI or ML against the conventional population PK model, common statistical metrics used for the external validation of any pharmacokinetic method are applied. These metrics evaluate mainly the accuracy or bias and the precision of the models. There is no consensus on how to appropriately design and interpret external evaluation of pharmacokinetic models. The section dedicated to “model validation” in the FDA’s guidelines for industry on population PK does not contain clear criteria on what tests should be performed, what metrics should be applied, or how to interpret them correctly [55]. Furthermore, some guidelines for reporting population PK studies do not consider the issue of external evaluation at all [56,57]. The absence of clear recommendations on how to carry out external validation and comparison of population methods has triggered wide variability in terms of the statistical parameters calculated in the different studies. Therefore, there are significant differences in study design, statistical methods, and reporting, making it very difficult to compare studies and interpret pooled results [52]. This problem is so serious that, in the articles analyzed in this review, we have found several inconsistencies regarding the name or abbreviation used for the statistical metrics used, as well as in the formula used for each of them, an aspect already noted by Meng et al. [58].

With the aim of clarifying this aspect and facilitating the understanding and interpretation of the results obtained in the articles collected in this review, Table 2 shows all the statistical metrics used in the articles analyzed, as well as their possible abbreviations, formulas, definitions, and interpretations. There are different ways to measure errors [59,60]. The simplest is to measure it like a difference. Within this type are the prediction error (PE) and the absolute prediction error (AE), which is the same as the previous one but in terms of absolute value. Another widely used possibility is to use square errors (SPEs) or root square errors (RMSEs). Squaring the error gives higher weight to the outliers, which results in a smooth gradient for small errors. Another way to express them is by calculating the mean (ME, MAE, MSE) or the median (MDE, MDAE) of all the errors in the sample. These errors present units of concentration (squared concentration units in the case of SPE or MSE) and the desired value would be the minimum possible. Another way to express the error is to express it relative to the range of the target variable (MRE, RMPE, RRMSE). These types of expressions limit the values between 0 and 1, with values closer to 1 being better. Most of these errors can also be expressed as a percentage (PE%, AE%, ME%, or MPE, MAE% or MAPE, MDPE%, MDAPE%, MRE%, RMSE%…), presenting values between 0 and 100. Some of these metrics are more focused on the measurement of accuracy or bias and others on precision, as expressed in Figure 3.

Likewise, there is no consensus regarding the model acceptability criteria once external validation has been carried out. Across all published studies, there is consensus that models accept bias thresholds between −20% and 20% and imprecision thresholds of <30% [61,62]. However, it is not uncommon for each article to use a different model acceptance range or for these results to not be expressed as a percentage, which in turn makes comparisons between publications very difficult [52].

## 3. Discussion

The findings of this review reveal that the application of pharmacokinetics for antibiotics with artificial intelligence and machine learning is very promising but, in its infancy, there is no regulatory framework to facilitate its proper clinical implementation. This is consistent with other reviews on the use of AI techniques in healthcare [63,64]. Population models have traditionally been used for the pharmacokinetics of antibiotics. The AI models seem to present several advantages compared to the traditional population models, particularly speed and capacity with ample data. Nevertheless, at this juncture, evidence suggests that these models do not consistently outperform conventional approaches in terms of accuracy and clinical feasibility.

One of the key features of AI is its ability to incorporate many clinical variables, such as the inclusion of specific biomarkers or patient conditions impacting PK [35]. This could be of help in complex populations, such as pediatric or critically ill patients, for whom the traditional models are of limited use. However, we must note that many of the studies included in this review are retrospective and may thus suffer from inherent biases and may be less generalizable to wider clinical populations. In addition, variability among chosen AI techniques and evaluation metrics further complicates the comparison between studies and strongly underscores such a necessity in this area. Currently, there is no consensus on the metrics used to compare these models with conventional PK methods. Since there is no standard, it becomes impossible to conduct a general examination of the outputs received, thereby underscoring the urgency for framed guidelines that would facilitate a consistent review of the precision and feasibility of AI models in practice. This lack of a proper set of guidelines makes it impossible for researchers to check and correlate results, thereby obstructing further application in clinical settings.

Despite these challenges, the prospects for AI in PK are bright [4,16,34]. In particular, in the field of antibiotics, fast progress of these techniques may support the building of predictive models for newly developed antibiotics which require strict monitoring as a function of antimicrobial resistance. The incorporation of AI with mechanistic approaches like PBPK models therefore provides a promising opportunity for enhancing the precision and clinical relevance of PK models. Thus, the promising results of AI techniques and their combination with classical PK methods may be auspicious; however, more consistent studies using similar evaluation metrics are needed before their general application in antibiotic therapeutic monitoring. At least for the time being, such algorithms need to be considered as adjuncts that will help clinicians and pharmacokinetic experts to make decisions rather than alternatives to the existing methods.

This review has several limitations. First, it is important to highlight the limited number of articles selected after applying the selection criteria proposed in the review, which may limit the capacity to analyze the subject. However, narrowing the selection criteria allows us to discard articles that do not provide relevant information on the proposed objectives. Second, as already mentioned, the enormous heterogeneity in the studies analyzed makes it difficult to draw conclusions from their joint analysis. Each study uses different AI approaches, such as neural networks, decision trees, and boosting methods, which complicates the direct comparison of the results. Although it is understandable, due to the emerging use of these techniques, that different approaches are taken in the studies, it would be advisable to standardize the way they are carried out in the future. Similarly, the use of different techniques for measuring precision and accuracy error and, in many cases, even the absence of the difference between these two measures makes the joint interpretation of the results extremely difficult. Although it is correct to use any of the techniques and metrics, it is important to highlight that their differences make it challenging to compare and analyze the results as a whole. Another significant limitation is that most of the included studies have a retrospective design, using large previously collected databases or a prospective design based on automated patient simulations. These approaches entail population selection biases and limit the translation of the results to the real population, where conditions can vary considerably. The use of simulations is commonplace in this field due to the need for large amounts of data to train artificial intelligence. It is true that the use of simulations can be limiting in terms of the complexity of the simulation of complex biological systems; however, it also offers striking advantages such as the possibility of covering many more clinical scenarios than with real data or reducing the reliance on extensive real-world datasets. Therefore, we do not believe that the use of simulations is in itself a limitation, but it is something that should be taken into account in order to compare the results obtained. Furthermore, the lack of external validations in different cohorts, which is not present in most of the studies, is noted. On the other hand, despite the potential usefulness of these techniques for the development of pharmacokinetic models of new antibiotics, it is important to underline that the studies analyzed focus on conventional antibiotics, highlighting, once again, the emerging state of this field of research. It is important to note the lack of explanation of the type of AI technique and its way of working in most of the studies analyzed, which undermines confidence in the results obtained. Finally, due to the use of a scoping review approach [65], no quality assessment was performed on the included studies.

## 4. Material and Methods

This study is based on the guidance framework for conducting scoping reviews in accordance with the Preferred Reporting Items for Systematic Review and Meta-Analysis (PRISMA) extension for scoping reviews (PRISMA-ScR) [65]. The project was preregistered on the OSF database, https://doi.org/10.17605/OSF.IO/5KPYB, accessed on 5 November 2024. A comprehensive search strategy was employed to gather relevant data from multiple databases. The reference lists of all included studies were manually reviewed to identify other relevant papers. This section details the databases searched, the time frame of the searches, and the specific search terms used.

### 4.1. Objectives

The objective of this review is to analyze the studies published to date on PK or PK/PD modeling for antibiotic dose optimization using AI techniques. In addition, this review aims to evaluate in detail the different AI techniques used and the statistical metrics applied to assess the accuracy of the prediction and its comparison with the conventional monitoring techniques.

### 4.2. Search Strategy

The EMBASE and MEDLINE (by PubMed and OVID) databases were searched from inception to January 2024 using the following terms: (“Artificial Intelligence” OR “Machine Learning” OR “neural network”) AND “pharmacokinetics” AND antibiotics.

Reference lists of relevant studies were analyzed for additional studies in the literature. Furthermore, after the analysis of the included studies, specific searches were carried out to define and interpret the AI techniques used and the statistical parameters used for the evaluation of prediction accuracy.

### 4.3. Eligibility Criteria

All articles that met the following inclusion criteria were included in the review: (1) use of some artificial intelligence technique to predict concentrations of antibiotics. The following studies were excluded: (1) articles that use AI for other aspects than PK; (2) papers that use AI for PK of non-antibiotics drugs; (3) articles not written in English or no full text available; and (4) studies in which the accuracy of the prediction with the technique used has not been evaluated (See Figure 1).

### 4.4. Data Extraction

Screening was carried out independently by two reviewers (IVR and EBV), with full texts of potentially relevant articles considered for inclusion. Disagreements between reviewers were resolved by consensus or with a third reviewer (FJTC). Study descriptors were extracted, including first author, year of publication, antibiotic analyzed, number of patients or simulations, participant characteristics, study objective, AI technique used for modeling software, clinical parameters involved, PK model with which it is compared, quality assessment on prediction accuracy, and the results and conclusions of the study.

### 4.5. Synthesis of Results

A narrative synthesis was produced for the included studies. As recommended in the methodology for scoping reviews, no formal assessment of the risk of bias in the studies was pursued [65]. Identification of gaps was accomplished through a review of areas covered by the studies following the aspects of care considered. The research team reviewed the study findings and formulated questions that would be useful in addressing the identified gaps.

## 5. Conclusions and Future Perspectives

The use of AI, particularly ML techniques, in PK has shown growing interest in recent years. Its application is of special interest in the PK of antibiotics due to two main factors: firstly, the reduction in time in the development of models, which would allow for the monitoring of new antibiotics that are being discovered, and secondly, the possibility of introducing more clinical variables that may affect the scope of the pharmacokinetic objective.

This review shows the current paradigm for using these techniques for the pharmacokinetic monitoring of antibiotics. Mainly, these are retrospective studies on large databases in which new algorithms developed by ML-based models such as XGBoost, neural networks, or decision trees, among others, are tested and compared to population PK. No studies have been found that use them for therapeutic drug monitoring of newly developed antibiotics.

The current reality is that we are still at a fairly preliminary point in the use of these techniques. Most of these models are inherently complex and lack explanations of the decision-making process, which makes their clinical application difficult. The design and methodology of the studies carried out, as well as the metrics used to evaluate their precision, are highly variable. It would be necessary to standardize the comparison metrics against the standard pharmacokinetic models. Therefore, although the results are promising, more robust studies are needed to be able to say that these techniques represent a revolution in the clinical PK of antibiotics. For now, AI algorithms appear to be complementary to standard population PK approaches.

## Figures and Tables

**Figure 1 antibiotics-13-01203-f001:**
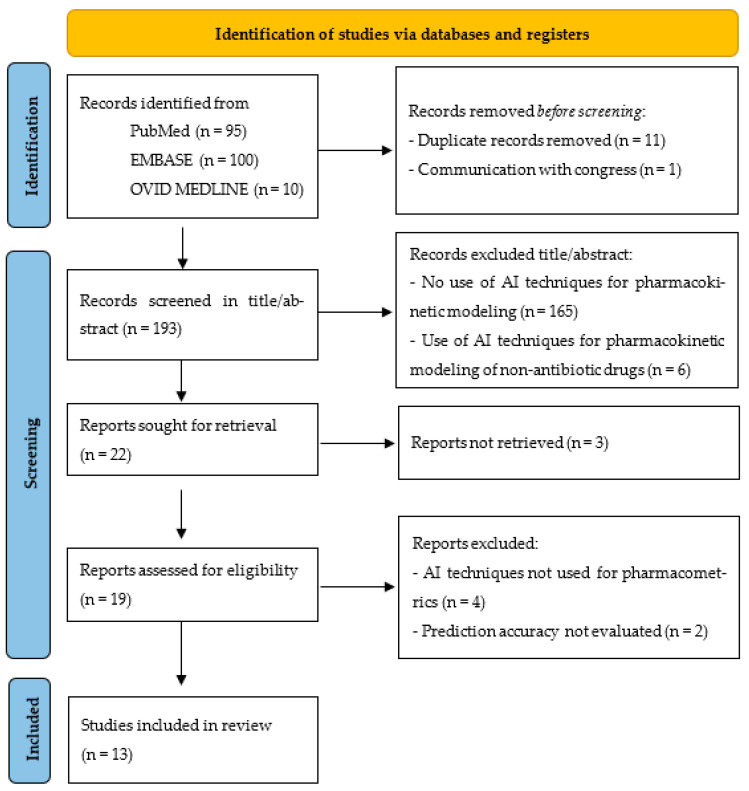
PRISMA flow diagram of search results and selection process of the studies. AI: artificial intelligence.

**Figure 2 antibiotics-13-01203-f002:**
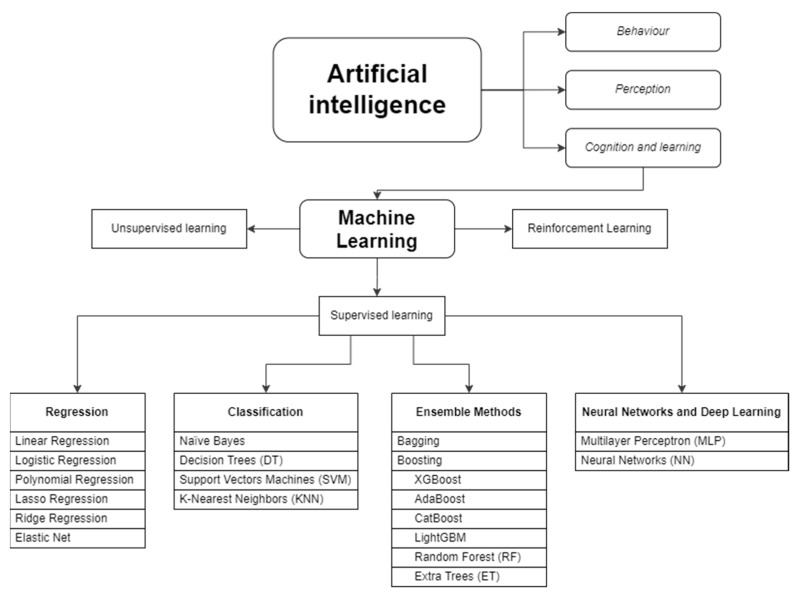
Artificial intelligence and machine learning technique classification.

**Figure 3 antibiotics-13-01203-f003:**
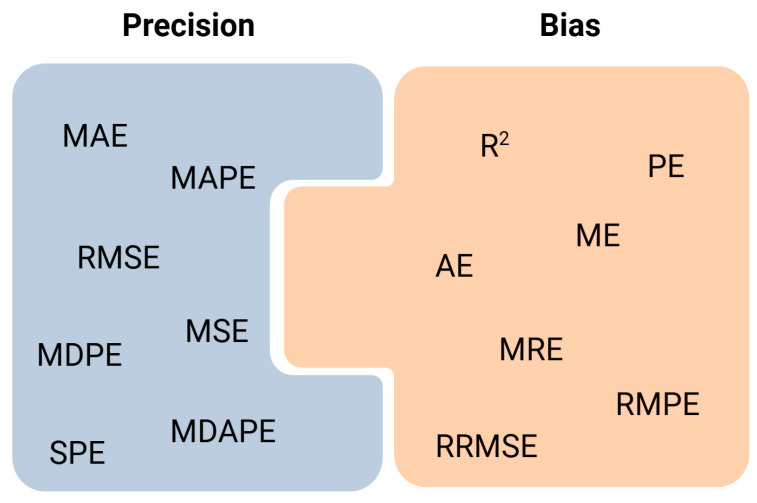
Classification of the main metrics for prediction accuracy measurement.

## Data Availability

No new data were created or analyzed in this study.

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
