# Peer review of "Artificial Intelligence and Machine Learning Applications to Pharmacokinetic Modeling and Dose Prediction of Antibiotics: A Scoping Review"

_antibiotics, 2024, doi:10.3390/antibiotics13121203_

Round 1
Reviewer 1 Report
Comments and Suggestions for Authors
The present article is a review of works in which AI, and above all machine learning (ML), was used to predict pharmacological/pharmacokinetic parameters in antibiotic therapy. The article is interesting, although in the final form it concerns only 13 papers. Initially, the authors analyzed over 200 of them, but most were excluded because they did not meet the assumptions adopted in this work. For this reason, the review may seem rather limited. On the other hand, these few works show how much researchers from different teams differ when using AI methods in their specific models. In my opinion, the greatest value of this review is to point a demand for the entire medical community to work out respective standard procedures for such purposes.
As I wrote above, the paper is interesting, but some issues should be improved.
I missed a broader analysis for papers where most of the data for ML was simulated. Can we refer to their effectiveness, compared to methods where real data was analyzed?
I also missed a broader discussion of the noticed imperfections in individual models. The authors of the present review write, among other things, about improper parameters for assessing precision, but do not elaborate on this problem. We do not know which parameters were incorrectly used and what effect it had on the obtained results.
The authors also write about the main idea of using AI to accelerate procedures related to the introduction of new antibiotics into therapy. Does the literature not include works devoted to such new antibiotics and the use of AI to assess their parameters?
Author Response
Reviewer 1:
The present article is a review of works in which AI, and above all machine learning (ML), was used to predict pharmacological/pharmacokinetic parameters in antibi-otic therapy. The article is interesting, although in the final form it concerns only 13 papers. Initially, the authors analyzed over 200 of them, but most were excluded because they did not meet the assumptions adopted in this work. For this reason, the review may seem rather limited. On the other hand, these few works show how much researchers from different teams differ when using AI methods in their specif-ic models. In my opinion, the greatest value of this review is to point a demand for the entire medical community to work out respective standard procedures for such purposes.
As I wrote above, the paper is interesting, but some issues should be improved.
I missed a broader analysis for papers where most of the data for ML was simulated. Can we refer to their effectiveness, compared to methods where real data was analyzed?
I also missed a broader discussion of the noticed imperfections in individual models. The authors of the present review write, among other things, about improper pa-rameters for assessing precision, but do not elaborate on this problem. We do not know which parameters were incorrectly used and what effect it had on the ob-tained results.
The authors also write about the main idea of using AI to accelerate procedures related to the introduction of new antibiotics into therapy. Does the literature not include works devoted to such new antibiotics and the use of AI to assess their parameters?
We sincerely thank you for your thoughtful and constructive comments, which have provided valuable insights to improve our manuscript. Below, we address each of your concerns:
- Articles analyzed: We acknowledge that our final selection of 13 papers may appear limited. However, we adopted strict inclusion criteria to ensure that the selected studies met the specific assumptions of this work. To address this, we have added a statement in the Discussion section acknowledging this limitation and emphasizing that our intention was to maintain the methodological rigor of our analysis. Additionally, we have emphasized the diversity of AI approaches and the need for standard procedures, as you have rightly noted as a key contribution of this review.
- Analysis of simulated vs. real data: Thank you for pointing out the need for a broader discussion on this topic. We have expanded the analysis in the Discussion section to discuss the effectiveness of models based on simulated data compared to those using real-world data, highlighting the advantages and limitations of each approach.
- Discussion of model imperfections: We really appreciate your input. We mentioned in the discussion section the important limitation on the wide variability of AI techniques used in the selected articles and the large variability of metrics used to measure the accuracy and precision of the methods. Although it is correct to use any of the techniques and metrics, we want to emphasize in the Discussion section that their differences make it challenging to compare and analyze the results as a whole. To address this, we have revised the Discussion section to clarify this point.
- AI and new antibiotics: Thank you for this point. As discussed in the Discussion section, we believe that the use of AI techniques could be used to expedite the development of pharmacokinetic models for new antibiotics, facilitating optimal dosing strategies from the outset. However, as highlighted in our review, this field is still in its early stages. Notably, these techniques have primarily been applied to widely used antibiotics, in which their pharmacokinetics are already known and there are already pharmacokinetic models developed by conventional techniques to compare the results. We have not found articles that address the objectives of this review with new antibiotics.

Reviewer 2 Report
Comments and Suggestions for Authors
There are a few typographical and spelling/grammatical errors which should be corrected:
Abstract: line 28 remove hyphen from anti-biotics, line 29 "for dose prediction" rather than "to dose prediction"
Introduction: line 60 "relationships" rather than "significant correlations"
Figure 1: "title" rather than "tittle", "techniques" rather than "technics"
Section 2.2: line 106 "authors" rather than "they"
Section 2.3: line 183 "selected" rather than "split"
Section 3: lines 318//319 repetition of highlight/highlighting
Section 4.1: line 334 "PK or PK/PD" rather than "PK/PD"
Author Response
Reviewer 2:
There are a few typographical and spelling/grammatical errors which should be corrected:
Abstract: line 28 remove hyphen from anti-biotics, line 29 "for dose prediction" ra-ther than "to dose prediction"
Introduction: line 60 "relationships" rather than "significant correlations"
Figure 1: "title" rather than "tittle", "techniques" rather than "technics"
Section 2.2: line 106 "authors" rather than "they"
Section 2.3: line 183 "selected" rather than "split"
Section 3: lines 318//319 repetition of highlight/highlighting
Section 4.1: line 334 "PK or PK/PD" rather than "PK/PD"
Thank you for the observations. We had changed all the errors.

Reviewer 3 Report
Comments and Suggestions for Authors
This is a good summary of the application of AI and machine learning in the field of antibiotic modeling and simulation. This review will contribute to further advances in the application of AI and machine learning in this field.
Author Response
Reviewer 3:
This is a good summary of the application of AI and machine learning in the field of antibiotic modeling and simulation. This review will contribute to further advances in the application of AI and machine learning in this field.
Thank you for your kind words and positive feedback on our review. We are delighted that you found the manuscript useful in this field. It is our hope that this work will indeed inspire further advancements and foster collaboration across the medical and AI research communities.